# Advances in Cellular Reprogramming-Based Approaches for Heart Regenerative Repair

**DOI:** 10.3390/cells11233914

**Published:** 2022-12-03

**Authors:** Xingyu He, Jialiang Liang, Christian Paul, Wei Huang, Suchandrima Dutta, Yigang Wang

**Affiliations:** 1Department of Pathology & Laboratory Medicine, College of Medicine, University of Cincinnati, Cincinnati, OH 45221, USA; 2Department of Internal Medicine, College of Medicine, University of Cincinnati, Cincinnati, OH 45221, USA

**Keywords:** stem cells, iPSC-CMs, engineered heart tissue, direct reprogramming, progenitor cells, regenerative heart repair, immune reduction, myocadiac infarction

## Abstract

Continuous loss of cardiomyocytes (CMs) is one of the fundamental characteristics of many heart diseases, which eventually can lead to heart failure. Due to the limited proliferation ability of human adult CMs, treatment efficacy has been limited in terms of fully repairing damaged hearts. It has been shown that cell lineage conversion can be achieved by using cell reprogramming approaches, including human induced pluripotent stem cells (hiPSCs), providing a promising therapeutic for regenerative heart medicine. Recent studies using advanced cellular reprogramming-based techniques have also contributed some new strategies for regenerative heart repair. In this review, hiPSC-derived cell therapeutic methods are introduced, and the clinical setting challenges (maturation, engraftment, immune response, scalability, and tumorigenicity), with potential solutions, are discussed. Inspired by the iPSC reprogramming, the approaches of direct cell lineage conversion are merging, such as induced cardiomyocyte-like cells (iCMs) and induced cardiac progenitor cells (iCPCs) derived from fibroblasts, without induction of pluripotency. The studies of cellular and molecular pathways also reveal that epigenetic resetting is the essential mechanism of reprogramming and lineage conversion. Therefore, CRISPR techniques that can be repurposed for genomic or epigenetic editing become attractive approaches for cellular reprogramming. In addition, viral and non-viral delivery strategies that are utilized to achieve CM reprogramming will be introduced, and the therapeutic effects of iCMs or iCPCs on myocardial infarction will be compared. After the improvement of reprogramming efficiency by developing new techniques, reprogrammed iCPCs or iCMs will provide an alternative to hiPSC-based approaches for regenerative heart therapies, heart disease modeling, and new drug screening.

## 1. Introduction

Heart disease is the leading cause of death globally [1]. Due to the lack of advanced medical equipment and preventive knowledge, the morbidity and mortality rates in third-world countries are increasing significantly when compared to developed countries. Current research has provided a variety of pharmaceutical and surgical treatment methodologies to provide a high quality of life for patients. However, according to a study by Roth et al., the number of deaths due to heart disease has steadily increased from 1990 to 2019 [2]. In addition, disability-adjusted life years have also trended up during the last two decades.

One of the main barriers to heart disease treatment efficacy is the low regenerative ability (<1%) of human adult cardiomyocytes (CMs) [3]. CMs are heart muscle cells that provide heart contractile force and play an important role in whole-body metabolism through blood circulation [4]. Despite advances in pharmaceutical treatment, ischemic heart diseases often cause continuous loss of CMs due to the chronic oxygen and nutrient supplement shortage, which has been identified as the essential process of heart failure [5,6]. Thus, to reduce the increasing mortality rate, innovative treatment strategies are urgently needed for heart function recovery.

In recent studies, cell therapy has attracted attention, and several promising clinical trials have been completed in a variety of organs, including the liver and kidney [7,8]. The development of stem cell therapeutics provides opportunities for the treatment of heart diseases by transplanting functional cardiac cells into the damaged heart area [9]. Several researchers have reported that both human embryonic stem cells (hESCs) and human induced pluripotent cells (hiPSCs) can be derived into functional CMs in vitro [10,11]. Compared with hESC, hiPSCs have been considered the better candidate for cardiac cell derivation, because they avoid several ethical challenges. iPSCs can also be induced from a variety of cell types from the same patient for personalized medicine [12,13]. In addition, hiPSC-derived CMs show a better capacity for slow maturation, which contributes to an unlimited cardiac cell source for cell-based regenerative therapy [10]. In addition, the use of an autologous cell source could potentially avoid acute immune rejection [14]. However, there are still major challenges to optimizing the usage of hiPSC-derived cardiac cells. Significant challenges must be addressed in maturation, engraftment, immunogenicity, and scalability. Encouragingly, several practical strategies have been implemented by researchers to address these challenges. Concurrently, long-term therapeutic studies are necessary to evaluate the safety and efficiency of using hiPSC-derived functional cardiac cells.

In addition to iPSC-derived cell therapy, direct reprogramming techniques have gained significant attention in recent studies. Direct reprogramming techniques can be classified into virus-based and non-viral methods that can be used to directly convert cardiac fibroblasts into iCMs or cardiac progenitor cells (CPCs) [15], providing an alternative cell resource for heart regenerative medicine and cardiac repair. In this review, we examine the recent advanced strategies for cardiac regenerative approaches, which involve transplanting iPSC-derived cardiac cells. We also discuss the development of direct cardiac reprogramming techniques. Major technical challenges with the current enhanced methodologies through the most recent investigations will be included and discussed. The recent advanced methodologies for reprogramming based regenerative heart disease treatment schematic are demonstrated in Figure 1.

## 2. iPSC-Derived Cardiac Cell Therapy

hiPSC was first described by Japanese researcher Shinya Yamanaka, who has successfully reported induced pluripotent stem cells from human fibroblasts by four defined transcription factors: Oct3/4, Sox2, Klf4, and c-Myc [16]. The following researchers also demonstrated iPSC generation from skin or blood cells. With the development of the stem cell differentiation process, iPSCs can be further derived into different cell types including cardiomyocytes. The iPSC-derived cardiomyocytes revealed a great potential future for patient-specific treatment and regenerative therapy for cardiovascular diseases.

### Cardiac Cell Production

iPSC-derived cardiovascular cells represent tremendous possibilities for the regenerative heart repair research field. Among iPSC-derived cardiovascular cells, iPSC-CMs have demonstrated the most contributions to the functional improvement of the damaged heart [17]. Several optimized protocols have been published, with relatively high cardiomyocyte derivation efficiencies and functional biological characteristics. Monolayer-based differentiation protocols, inductive coculture protocols, and spin-embryoid body (Spin-EB) protocols are the three major approaches for deriving iPSCs into CMs [18,19,20]. In addition, CM differentiation can also be achieved through small molecules by tuning signal pathways of heart development, as summarized in Figure 2 below [21].

The cardiac differentiation process is based on three stages through spatial-temporal modulation signaling pathways [22]. At the early stage of differentiation, Sean Wu’s group found that bioactive lipids such as sphingosine-1-phosphate (S1P) and lysophosphatidic acid (LPA) can independently enhance CM generation, which increased the nuclear accumulation of β-catenin and Wnt signaling pathway mediators [21]. Activation of several major signaling pathways (BMPs, Wnts, TGF-β/Activin/Nodal, and FGFs) can contribute to a high yield and purity of iPSC-CMs [23]. The successful application of combinations of different growth factors and inhibitors demonstrates the benefits of synergistic induction for cardiac gene activation [24].

In addition, researchers have also investigated the subtype-directed differentiation of hiPSCs into atrial and ventricular CMs. Lenz et al. reported a method that could derive iPSCs into atrial and ventricular CMs in feeder-free conditions [25]. In their protocol, retinoic acid played an important role in tuning the population of atrial and ventricular CMs, while generally higher retinoic acid could produce more atrial CMs [26]. Atrial CMs usually have a stronger contraction force than ventricular CMs; therefore, generation of atrial CMs could provide better relief for severe heart failure symptoms.

## 3. Challenges for Usage of iPSC-Derived Cells in Heart Disease Therapy

In addition to the benefits mentioned above, there are existing challenges that limit the use of iPSC-based cell therapeutics in regenerative heart medicine. Recent publications have suggested that semi-maturation, low engraftment rate, a strong immune response, uncertain tumorigenicity, and limitable scalability are the major barriers to overcome for translational studies.

### 3.1. Maturation Enhancement of iPSC-CMs

iPSC-derived cardiac cells can provide a number of opportunities for cardiac regenerative treatment. However, the functional immaturity of iPSC-derived CMs remains a hurdle for translational clinical applications. iPSC-CMs are immature in metabolic signature, electrophysiological properties, and ultrastructure features [27] when compared with mature CMs. These properties are associated with other issues (such as arrhythmia and low engraftment rate), which are a function of unmatched electrophysiological characters with host cardiac cells and low expression of myocardial-specific proteins [28]. In addition, these immature iPSC-CMs demonstrate different metabolic pathways mainly based on glycolysis, while adult CMs synthesize ATP through fatty acid β-oxidation, which is a more efficient energy supply [29]. This inefficiency of energy generation further suppresses contractile force performance.

In recent research, the maturation process has gained attention for making immature iPSC-CMs more physiologically close to adult CMs, which leads to better therapeutic potential. Notably, co-culture of hiPSC- CMs with non-CMs (such as endothelial cells and mesenchymal stem cells (MSCs)) has been shown to promote maturation with cardiac gene expression enhancement and structural improvement [30]. The rationale for better maturation can be attributed to the presence of growth factors such as bFGF, VEGF, SDF-1, and GM-CSF, which are secreted by MSCs to mediate the iPSC-CMs maturating process and improve electrical coupling [30]. Similarly, a co-culture with endothelial cells could offer a better surrounding microenvironment by expressing extracellular matrices, which could increase the sarcomere length of iPSC-CMs [31].

In addition to the co-culture approach, electrical stimulation could accelerate the maturation process of iPSC-CMs. Wang’s group suggested that electrical stimulation for hiPSCs from 2 Hz to 6 Hz within 2 weeks may accelerate cardiac cell differentiation and enhance the maturation of hiPSC-CMs by showing adult-like structural gene expression [32]. Moreover, electrically stimulated hiPSC-CMs demonstrated better calcium ion handling and contraction force, with ultrastructure improvement.

Moreover, in terms of the chemical manipulation approach, recent investigators have indicated that inhibition of mTOR could promote hiPSC-CMs maturation [33]. They explained that transient treatment of human iPSC-derived CMs with Torin1 shifted cells to a quiescent state and enhanced cardiomyocyte maturity.

Three-dimensional engineered heart tissue (3D-EHT) has also been widely used for regenerative medicine to treat heart disease, with several advantages [34,35]. For example, by embedding iPSC-CMs in the 3D-EHT, Plakhotnik et al. studied the effect of different mechanical strain magnitudes on the maturation of iPSC-CMs [36]. They reported that the contractility of iPSC-CMs positively correlated with the strain magnitude, and plateaued at around the 15% strain. From a genetic level perspective, the expression of the beta-myosin heavy chain (MYH7) has been improved in these mechanically stimulated iPSC-CMs. It is beneficial that engineered heart tissue could also be combined with other methods, such as electrical stimulation, for cardiac function enhancement [37].

Despite having demonstrated improved iPSC-CMs maturation, there are still some physiological differences between iPSC-CMs and adult CMs. In order to efficiently generate mature hiPSC-CMs, fundamental mechanisms should be further investigated to reduce maturation time and production cost.

### 3.2. Engraftment Improvement of iPSC-CMs

Another challenge in utilizing iPSC-derived cardiac cells is a very low engraftment rate after cell transplantation. A low engraftment rate directly limits therapeutic efficacy through a lack of sufficient contraction force to improve decreasing heart functions. Wolfram et al. found that approximately 90% of cells were lost in their predesigned engineered heart tissue within 14 days after transplantation surgery [38]. Even with an improved approach that applied cycling mechanical stimulation, the engrafted cell rate remained less than 20% after 30 days of transplantation. Thus, it is crucial to understand the barriers to the engraftment rate in order to improve the survival of iPSC-CMs.

In terms of the delivery route, intravenous injection, intramyocardial injection, and transplantation of engineered heart tissue/sheets are the major approaches for utilizing iPSC-CMs [39]. Most of these approaches have demonstrated a certain level of heart function improvement under various heart disease models [20,40]. These approaches indicate a promising future for the field of regenerative cardiovascular disease treatment.

The ischemic heart disease region is associated with a harsh environment for recipient tissue and implanted iPSC-CMs. Although functional maturation can be enhanced using the various approaches discussed above, iPSC-CMs have increased susceptibility to hypoxia-induced damage [41]. Insufficient oxygen and nutrient supply lead to a significant iPSC-CM loss from days to weeks after transplantation, suggesting that enhancement of angiogenesis with local blood perfusion could promote the iPSC-CMs survival rate, and potentially improve cardiac functions [42,43].

While several strategies of angiogenesis enhancement have been explored by scientists, pre-designed microvessels in the engineered heart tissues have demonstrated significant improvement in the remuscularization of infarcted hearts in a rat model [44]. In these experiments, polydimethylsiloxane (PDMS) was used as a mold to design the microvessels, followed by the vascularization of endothelial cells, then the iPSC-CMs were engaged in the pre-designed microvessels. This pre-vascularized design could reduce the time for angiogenesis to occur between host tissue and transplanted engineered heart tissues.

Supplements of growth factors have also shown some advantages for engraftment enhancement [45]. Yamashita et al. demonstrated that using stage-specific supplementation of vascular endothelial cell growth factor (VEGF) significantly improved the cardiac function of more than 40% of hiPSC-CMs that remained alive after 4 weeks of transplantation in a myocardial infarction rat model [46]. This engraftment rate almost doubles Wolfram’s record after 30 days of transplantation. Similarly, engineered heart tissue containing hiPSC-CMs and hiPSC-ECs demonstrated a better engraftment rate and better therapeutic efficiency, while hiPSC-ECs provided strong therapeutic angiogenesis, which improved microvascular with blood circulation improvement [47]. The co-culture system in the engineered heart tissue could be modified with a bionic method, mimicking the complexity of cardiac tissue.

At the same time, in another recent study from Lei Ye’s group, a combination of using thymosin β4 (Tb4) microspheres and hiPSC-CMs was used to enhance the engraftment rate with reparative potency for myocardial repair in large animals [48]. They also demonstrated that there was no ventricular arrhythmia during their 4-week monitoring period.

It is rarely reported that embedding oxygen-release nano/micro-particles in engineered heart tissue could potentially provide another strategy for increasing the engraftment rate by improving the harsh microenvironment. Guan’s group has reported a hypoxia-sensitive system of oxygen-release microspheres for mesenchymal stem cell (MSC) survival support [49]. In their design, a core–shell structure microsphere was synthesized through coaxial electrospraying. Hydrogen peroxide (H2O2) was applied as an oxygen-generating agent with an oxygen-responsive shell which could reduce the possibility of reactive oxygen species (ROS) -induced apoptosis in the surrounding cells. The environment-responsible oxygen release promoted the cells’ survival rate after transplantation. This strategy could also be combined with other methods mentioned above to achieve a potentially better engraftment rate.

However, most of the achieved engraftment rate studies were about 4–8 weeks, and the long-term engraftment rate and survival rate of transplanted iPSC-CMs studies are still lacking. It is concerning that the immune response between implanted derived cells or engineered tissue with original host tissues could also significantly affect the long-term engraftment rate and biosafety. Additional innovative strategies are also required for large animal therapeutic efficacy and safety evaluation.

### 3.3. Immune Response to hiPSC-CMs

Generation of hiPSC-CM for autologous cell transplantation is unfeasible in the clinical setting due to time restrictions and high costs. Therefore, the generation of allogeneic iPSC-CMs as ‘off-shelf’ productions or deposited in a biobank becomes an attractive option for patients with heart failure. However, immune rejection of allogeneic iPSC-CMs is a significant concern. Many studies have indicated that transplanted iPSC-CMs might not be widely accepted by the recipient’s immune system [50]. Evidence has shown that transplantation of genetically dissimilar iPSCs-derived cells, even within species (allogeneic), can induce immune rejection with activated macrophages accumulating in the transplanted iPSC-CMs area [51]. It has been considered that this immune rejection might be limited by matching the major histocompatibility complex (MHC) antigens between the donor and the recipient [50]. Currently, it is reported that MHC-matched iPSC-CMs survive in myocardial infarcted monkeys with no evidence of immune rejection for 12 weeks [52].

The human leukocyte antigen (HLA) gene is considered one of the key factors for the immune system to recognize self- or non-self-components. It, therefore, plays an important role in the field of immune reduction research for transplantation-based treatments. There are several major strategies to reduce immune rejection of transplanted iPSC-CMs through matching HLA or removing HLA of transplanted cells [53,54]. HLA class I molecules (such as HLA-C and HLA-G), PD-L1, and CD47 are known as immune tolerance-related factors of iPSCs. “Universal” or “hypoimmunogenic” hiPSCs were designed through CRISPR/Cas9 gene-editing techniques by knocking out these immune-related genes/factors, which could effectively enhance immune compatibility [55]. Moreover, these gene-edited “universal“ cells could efficiently escape activation of T cells, NK cells, and macrophages.

In addition, syngeneic mesenchymal stem cell (MSC) co-transplantation might thus reduce allogeneic iPSC-CM rejection by mediating immune tolerance via regulatory T cells, as well as cell–cell contact with activated lymphocytes [56]. It was further explained that the MSC co-transplantation increased CD4+CD25+FOXP3+regulatory T cell numbers, apoptotic CD8-positive T cells, and IL-10 and TGF-beta expression at the implantation site. These approaches have promise for cardiomyogenesis-based therapy using allogeneic iPSC-CMs for severe heart failure [57].

### 3.4. Tumorigenicity of hiPSCs Derived Cardiac Cells

Safety concerns are the primary priority for clinical applications that apply the use of iPSC-derived cardiac cells. Recent studies have shown that incompletely differentiated hiPSCs could form malignant tumors after transplantation in in vivo application [58]. However, various studies have demonstrated their protocols, and that high purity (>95%) and functional hiPSC-CMs could be generated by tuning the signaling and metabolism pathway [59,60]. However, in reality, even if the ratio of undifferentiated iPSCs is less than 0.3%, these iPSCs can still lead to tumor formation, which has been tested in rats [61]. The proliferation ability and tumor-related gene expressions or mutations in the iPSC could potentially be explained as the main causes of tumorigenicity [62,63]. Therefore, the detection of undifferentiated and tumorigenic iPSCs is critical to make hiPSCs safely usable for the desired clinical applications. Wang’s group has developed an ultrasensitive and rapid quantification method to determine the rare tumorigenic stem cells in the hiPSC-derived cardiomyocyte population [64]. Based on the tumor detection strategies, their stem cell quantitative cytometry (SCQC) system has a sensitivity that can determine underived and rare tumorigenic hiPSC, as low as 0.0005%, in populations of hiPSC-CMs. Their technique provides a proper platform to develop anti-tumor approaches for eliminating the possibility of tumorigenicity related to underived iPSCs after in vivo transplantation.

In addition, anti-tumor drugs such as doxorubicin (at a non-cardiotoxic level) could be applied as purifying agents, selectively killing the rapidly proliferating cells, such as underived hiPSCs, without affecting the normal derived cardiac cells [65]. These purification strategies could significantly reduce the tumorigenicity due to undifferentiated hiPSCs or other stem cells. In general, the combination of ultrasensitive tumorigenic detection and anti-tumor purification strategies demonstrates great practical methods for reducing the risk of tumorigenicity of hiPSCs and hiPSC-derived cells.

Studies on the long-term tumorigenicity of transplanted hiPSC-derived cardiovascular cells are still significantly lacking. Due to the potential for incomplete reprogramming and accumulated DNA damage of iPSC-derived cardiac cells, safety must be the highest priority before moving forward into clinical trials.

### 3.5. Scalability Expansion of iPSC-CMs

It is reported that the high fatality of heart diseases such as myocardial infarction (MI) could result in approximately 1 billion cardiomyocyte losses in the infarct border zone [66]. In order to properly restore heart function and prevent further heart damage, a large population of mature iPSC-CMs and other supporting cardiovascular cells are required for the replacement of the contractile units in the damaged area. Therefore, scalability is crucial for regenerative heart disease treatment involving transplantation approaches. In addition, cost and cell culture time should also be taken into consideration to make the treatment more practical and affordable. Encouragingly, several new methods have been developed for a large scale of relatively mature iPSC-CMs production.

A large-scale mature hiPSC-CMs generation method was reported through the aggregation differentiation protocol [67]. Briefly, aggregated hiPSC were treated by BMP4, activin A, and Wnt inhibitors in StemPro medium in sequence. Then, differentiating aggregates were dissociated into single cells in VEGF StemPro medium, followed by RPMI-1640 medium supplemented with insulin-containing B-27 supplement. They clarified that they could generate more than 108 mature hPSC-CMs economically and efficiently in PDMS-roller bottles.

At the same time, Keiichi Fukuda et al. demonstrated the production of mouse and human pluripotent stem cell-derived CMs through a metabolic flow method [68]. Since they found that only derived CMs could survive in the glucose-depleted and lactate-enriched environment during the cell culture process, eventually, by this method, they could reach approximately 99% purity of CMs. They also mentioned that there was no tumor formation after 2 months of transplantation with purified iPSC-CMs.

Overall, there are many complex interconnections among maturation, engraftment rate, and immune response of using iPSC- derived CMs and other cardiac cells which must be addressed before moving forward into clinical applications.

## 4. Cardiac Reprogramming for Heart Regeneration

Non-CMs also play important roles in regular heart functions such as physiological support and tissue remodeling. In terms of the ratio of cell populations, most of these non-CMs are cardiac fibroblasts [69]. In an injured heart, cardiac fibroblasts could be converted into myofibroblasts, the process of which will further contribute to cardiac fibrosis [70]. Inspired by the iPSC-differentiation approaches, scientists found that cardiac fibroblasts can be directly reprogrammed into cardiomyocyte-like cells (iCMs) through transgenic techniques without passing through a stem cell state or a pluripotent stage [71]. Transcription factors, including Gata4, Mefc2, and Tbx5, were first found to successfully reprogram myofibroblasts into induced cardiomyocytes. After that, microRNAs and small molecules showed great potential to further improve the reprogramming efficiency. This direct-reprogramming strategy provides an attractive concept for generating functional cardiac cells in the injured heart area for the purpose of mitigating patients’ symptoms. It could potentially reduce the chance of cardiac fibrosis and restore heart function by generating induced CMs (iCMs) and induced cardiac progenitor cells (iCPCs) [72,73]. Direct reprogramming methodologies could significantly lower the tumor formation risk, based on transcriptional factors such as genetic or small chemical molecules that avoid the pluripotent state. In addition, because of the wider choice of cell sources, the reprogramming technique could further contribute to the development of autologous cell transplantation for heart disease treatment, which could reduce the concern of immune rejection, as was mentioned in the previous sections [74].

Moreover, based on the reprogramming reagents, we briefly summarized direct-reprogramming techniques, including virus-based and non-viral cardiogenic approaches. These approaches provide additional strategies for cardiovascular cell generation both in vitro and in vivo.

### 4.1. Virus-Based Cardiac Reprogramming

Scientists have been applying the transduction abilities of different viruses for cardiac reprogramming technology. Lentivirus, adeno-associated virus (AAVs), retrovirus, and sendal virus (Sev)-based vectors have been shown to have transgenic delivery abilities for cardiac reprogramming applications [75,76,77]. Kazutaka Miyamoto’s group has demonstrated their viral-based reprogramming approaches in induced cardiomyocyte-like cells (iCMs) for myocardial infarction treatment. Essential cardiac transcriptional factors (Gata4, Mef2, and Tbx5) play a crucial role in transcriptional regulation during embryogenesis and reprogramming. They have recently revealed that their Sev-GMT (Gata4, Mef2c, and Tbx5) transduction generated 100-fold more beating iCMs than retroviral-GMT, and shortened the duration to induce beating cells from 30 to 10 days in mouse fibroblasts [76]. In addition, it has been reported that chemical small molecules like SB431542 (TGF-β inhibitors) and XAV939 (WNT inhibitor) could enhance the efficiency, speed, and quality of iCM generation, both in vivo and in vitro, by downregulating fibroblast gene expression and activating cardiac gene expression [78].

Fibroblasts, T-lymphocytes, keratinocytes, and renal tubular cells have been successfully reprogrammed into induced CMs through the use of different reprogramming factors [79,80,81,82]. However, due to the sources of cell types, and in terms of the benefits of in situ reprogramming, cardiac fibroblasts have been considered an ideal candidate among these cells. It was reported that transcriptional factors, such as Gata4/Mef2c/Tbx5 (GMT), could directly reprogram fibroblasts into cardiomyocyte-like cells. Furthermore, a combination of these three factors could provide a rapid and efficient reprogramming approach for induced cardiomyocytes (iCMs) with spontaneous contraction and action potentials both in vitro and in vivo [83].

Lentiviruses and adeno-associated viruses (AAVs) have also been widely used in basic research studies for the past 10 years. Both lentiviruses and AAVs can deliver desired genes in both non-dividing and dividing cells with relatively long-term and stable expression [84]. Moreover, lentiviruses and most AAVs can infect any proliferating cells with low targeting ability. However, in clinical studies, AAVs are considered a better candidate, because lentiviruses are fundamentally integrated into the host genome. Due to the random infection nature of these viruses, they directly decrease the conversion efficiency of reprogramming and increase uncertainty for therapeutic applications. However, with the development of CRISPR/Cas9 technology, a combination of viral-based vector delivery systems and CRISPR/Cas9 systems could provide a target-specific and highly efficient strategy for in vivo transcriptional factor reprogramming [85]. It has been reported that the AAV-mediated CRISPR genome editing system has been used in many cardiovascular disease treatments with great progress [86]. In addition, encouragingly, AAV serotype1 was found to have selectivity towards cardiac fibroblasts, which could promote efficient reprogramming and potentially reduce the risk of insertional mutagenesis [87].

Overall, high transgene expression is critical for cardiac reprogramming efficiency, and safety concerns are still pushing researchers to create a better therapeutic regimen.

### 4.2. Non-Viral Cardiac Reprogramming

The majority of approaches to generate iCM use a virus-based delivery system for reprogramming factors. Correspondingly, these methods (as described above) have raised safety concerns, which include immunogenicity, insertional mutagenesis, and tumorigenicity. Non-viral direct reprogramming is a preferable method for clinical applications as it avoids the use of viruses.

In 2012, Dr. Dzau’s group first demonstrated that microRNA mediation could be used to directly reprogram cardiac fibroblasts into CMs in vitro [71]. With the development of cardiac signal pathways, microRNA transfection has become an ideal candidate for direct reprogramming. They showed that mRNAs (Gata4, Mef2c, and Tbx5 (GMT)) were employed to directly switch the cell fate of fibroblasts into iCMs, as shown by the expression of mature cardiomyocyte markers, sarcomere organization, and exhibition of spontaneous calcium flux characteristic of a cardiomyocyte-like phenotype. In addition, their following studies revealed that a combination of four microRNAs, 1, 133, 208, and 499, delivered by Dharmafect1 (a commercial lipid transfection product) demonstrated the highest reprogramming efficiency among their experimental groups [88]. However, for their in vivo experiments, the mRNAs were delivered through lentiviruses, which indicated the limitation of the current development of non-viral-based reprogramming choices and efficiency.

Similarly, to improve direct reprogramming efficiency, Niren et al. reported a targeting mRNA direct reprogramming method that converted mouse cardiac fibroblasts into cardiomyocyte-like cells [89]. The targeting ability was achieved through polyarginine-fused heart-targeting peptide and lipofectamine complex.

Nanoparticle gene delivery techniques provide an alternative method for cardiac reprogramming. It has been reported that mesoporous silicon nanoparticles (MSNs) coated with FH peptide-modified neutrophil-mimicking membranes were able to reprogram fibroblasts into iCMs, both in vitro and in vivo [90]. The rationale was based on the natural inflammation-homing ability of neutrophil membrane protein as well as FH peptide’s high affinity to tenascin-C (TN-C) produced by CFs. These MSNs could deliver microRNA1, 133, 208, and 499 (miR Combo) into CFs in mice through intravenous injection. These mRNAs regulate H3K27 methyltransferase and demethylase expression, which could promote iCM proliferation and the expression of contractile protein (MHC) [75].

In addition, Kim’s group has explored using cationic gold nanoparticles for the delivery of transcriptional genes such as Gata4, Mef2c, and Tbx5 for myocardial infarction treatment [91]. By conjugating cationic gold nanoparticles with polyethyleneimine (PEI), they have efficiently achieved a transition from cardiac fibroblast into iCMS, with heart function improvement.

Small chemical molecules can also be used to efficiently induce cardiac reprogramming [92,93,94]. PTC-209, for example, inhibits the expression of Bmi1, which is a critical barrier to iCM induction, through epigenetic modulation. A reduction in Bmi1 could increase active histone marker H3K4me3 and reduce repressive H2AK119ub [95]. As a result, suppression of Bmi1 could indirectly increase the expression of Gata4. It is reported that chemical molecules are more clinically amendable due to their adjustable dose and injection intervals. Huang’s group has shown that a chemical combination of CRFVPTM (C, CHIR99021; R, RepSox; F, Forskolin; V, VPA; P, Parnate; T, TTNPB; M, Rolipram) can induce the generation of iCMs from cardiac fibroblasts in normal adult mice in vivo [96]. Although not presented in detail in their publication, the functionality and mechanism for each chemical mentioned in their study are explained in greater detail here. CHIR99021 is a GSK-3 inhibitor that further contributes to Wnt signaling pathway activation and promotes cardiac mesoderm lineage commitment during differentiation. RepSox is an inhibitor of the transforming growth factor-beta receptor I (TGF-β) that reduces the interaction between JMJD3 with Gata4 [72]. TGF-β inhibitors are also able to increase GHMT-based cardiac reprogramming [97]. Forskolin is an adenylyl cyclase activator that increases the reprogramming efficiency dramatically by increasing the cellular concentration of cyclic AMP (cAMP) and cAMP-mediated functions [98]. Valproic acid (VPA) works as a chromatin remodeling enzyme inhibitor that promotes histone acetylation [99]. Parnate is a histone demethylase inhibitor that is used to regulate the histone acetylation and methylation with VPA for better reprogramming efficiency. In addition, TTNPB is a retinoid pathway activator that potently and selectively activates retinoic acid receptors (RARs). As mentioned in the previous section, retinoic acid signaling is critical for heart development and iPSC derivation in that it establishes anteroposterior polarity, formation of inflow and outflow tract progenitors, and growth of the ventricular compact wall [100]. Furthermore, rolipram (a phosphodiesterase (PDE) 4 inhibitor) can decrease the expression of inflammatory cytokines. After their 6-week experiment, the small-molecule cocktail group had reduced scar formation in their myocardial infarction model, and the heart ejection fraction (EF) was enhanced. However, they also noticed that their tdTomato+ cells were also found in the liver and lung without cardiomyocyte marker expression. A delivery vehicle loaded with small molecules for targeting the cardiac area could potentially reduce safety concerns, especially for long-term studies.

Kim’s group also revealed an ultra-efficient direct reprogramming method for converting fibroblasts into CMs using the extracellular vesicles (EV) [101]. In their study, mouse embryonic fibroblasts (MEFs) were treated with Emb-EVs (EVs present in the medium from the first stage of differentiation (EB (embryoid body) formation) for 10 days, then Emb-EVs were replaced with Mes-EV (stages before mesodermal induction are termed Emb-EV). According to their in vivo study, the effects on the EVs-cotreated group demonstrated great potential for myocardial infarction disease treatment, with approximately 60% reprogramming efficiency.

### 4.3. Reprogramming of Induced Cardiac Progenitors

Fibroblasts can also be reprogrammed into induced cardiac progenitor cells (iCPCs), similar to induced CMs, by overexpression of transcription factors. Induced CPCs can derive into major cardiovascular cell lineages. Thus, the production of iCPCs could not only generate CMs, but also endothelial cells (ECs) and smooth muscle cells (SMCs), which could be beneficial for the vascularization of the heart disease area [102]. Moreover, compared with induced cardiomyocytes (iCMs), induced cardiac progenitor cells are more scalable under proper signaling adjustment. Under chemically defined conditions, the iCPCs could proliferate more than 18 passages [103]. In addition, there are wider cell sources for iCPCs generation, such as human dermal fibroblasts, mouse tail-tip fibroblasts (TTFs), adult mouse lung fibroblasts (AL Fibs), and cardiac fibroblasts for both humans and mice [104].

Lalit et al. reported their research findings about the combination of 11 early cardiac factors (Mesp1, Mesp2, Gata4, Gata6, Baf60c, SRF, Isl1, Nkx2.5, Irx4, Tbx5, and Tbx20) to infect adult cardiac fibroblasts. They found that the combination of Mesp1, Tbx5, Gata4, Nkx2.5, and Baf60c (MTGNB) demonstrated a sufficient reprogramming ability for converting mouse cardiac fibroblast cells into iCPCs [73]. In addition, they also reported that Wnt and JAK/STAT signaling enables robust expansion of iCPCs. These induced cardiac progenitor cells could be further derived into CMs, endothelial cells, and smooth muscle cells for myocardial infarction treatment, both in vivo and in vitro.

Combined with CRISPR/Cas9-based transcriptional activators, many somatic fibroblasts could be reprogrammed into iCPCs [105]. Our group has recently reported that the activation of endogenous genes, such as Gata4, Nkx2.5, and Tbx5, can rapidly establish autoregulatory loops and initiate CPC generation in adult extracardiac fibroblasts using a CRISPR activation system [106]. Our transcriptomic analysis demonstrated that cell cycle and heart development pathways were important for accelerating CPC formation during the early reprogramming stage.

Recently, Li et al. reported high-efficiency protein transduction for reprogramming human dermal fibroblasts into cardiac progenitor cells [107]. Their high-efficiency protein (QQ-reagent) was able to deliver cardiac transcription factors into human dermal fibroblast cells within 6 h. By combining QQ-reagent-modified Gata4, Hand2, Mef2c, and Tbx5 with three cytokines, their protein-based factors’ delivery complexes could efficiently reprogram human dermal fibroblasts (HDFs) into iCPCs. According to the chromatin immunoprecipitation quantitative polymerase chain reaction assay, it was proven that this reprogramming process enhanced trimethylated histone H3 lysine 4, monoacetylated histone H3 lysine 9, and Baf60c at the Nkx2.5 cardiac enhancer region.

Similar to the iCM reprogramming process, small molecules could also contribute to iCPC generation [108]. The chemically-induced CPCs can be preserved into long-term proliferation. Further differentiation of iCPCs could be used for drug discovery, disease modeling, and cardiac cell therapy.

However, there are remaining physiological differences between native embryonic CPC development and reprogrammed iCPCs production. Further, the overall conversion efficiency of iCPCs remains low and requires improvement. In addition, safety concerns with long-term reliability need to be evaluated in different heart disease models.

### 4.4. Molecular Mechanisms of Cellular Reprogramming-Based Approaches

These cellular reprogramming studies have prompted a reassessment of restricted cell differentiation, which is delineated by Waddington’s epigenetic landscape model [109]. Converting somatic cells (e.g., fibroblasts) into the desired cell fate (e.g., CM) can be achieved by inducing pluripotency, intermediate progenitor state, or direct transdifferentiation (Figure 3A). Therefore, translating these reprogramming approaches into a clinical setting requires a greater understanding of the epigenetic mechanism underlying various processes of cell lineage conversion. The related mechanisms of iPSC or iCM reprogramming have been systemically discussed elsewhere [110,111,112,113]. Here, the potential mechanisms of iCPC reprogramming (Figure 3B) are summarized.

Although there are several different approaches for iCPC generation, they may share common pathways during the process of fibroblast induction. The transcriptomes of iCPCs generated from different reprogramming approaches have been profiled by next-generation sequencing, and the downstream analyses show the involved pathways, such as cell cycle, heart development, and Notch signaling [73]. The hallmarks of iCPC include expression of cardiac transcription factors (such as Nkx2.5, Gata4, and Isl1), cell surface markers (such as Flk1, Ssea1, and Cxcr4), cell renewal/proliferation, and cardiac tri-lineage potential [106]. Fibroblasts must overcome epigenetic barriers or eliminate the fibrogenic potential to acquire cardiac progenitor-like gene expression profiles and chromatin patterns. After epigenetic remodeling, various growth factors or small molecules can be used to enhance cell expansion and maintain stemness. Recent studies of iCPC reprogramming have brought new insights into the transcriptional and epigenetic mechanisms (Figure 3C).

Lineage-specific master regulators can serve as pioneer factors to bind and open closed chromatin with the binding of other transcription factors. We found that Nkx2.5, Gata4, and Tbx5 were essential for iCPC formation [106]. Interestingly, the co-occupancy of a chromatin region by the three factors has been revealed by chromatin immunoprecipitation and high-throughput sequencing (ChIPseq) in embryonic hearts [114], but it is worth revisiting their binding sites in the fibroblast genome in the setting of iCPC reprogramming by ChIPseq. The CRISPRa system can serve as a locus-specific activator to open the silenced chromatin locus that tightly represses cardiac gene expression in fibroblasts [105]. Studies on iPSCs or iCMs have also demonstrated that increasing chromatin accessibility using a pioneer factor can facilitate cell lineage reprogramming [112]. Therefore, Gata4, Nkx2.5, and Tbx5 can reinforce their own expression by directly binding their own promoter or enhancer elements in the induced fibroblasts.

In addition, the post-translational modifications of histones play a critical role in the chronic epigenetic stability of a cell state, which can avoid the reversion of the cells back to the initial fibroblast fate in a refractory manner. We found that the H3K4me3 (an active histone marker) levels were increased independently of the continuous expression of reprogramming factors [106]. Other histone modifications, such as H3K27me3 and H3K9ac, are also involved in the process of iCPC reprogramming. Therefore, the reprogramming factors may open heterochromatin, which allows the access of histone modifiers in the targeted loci (Figure 3C), as demonstrated in iPSC studies [112]. However, the molecular mechanisms of reprogramming factors recruiting the epigenetic writers remain unclear, and require further investigation in the setting of iCPC reprogramming. Moreover, other epigenetic mechanisms, such as DNA methylation, histone variants, and non-coding RNAs, may be involved in iCPC reprogramming, and also require further investigation with high-throughput sequencing approaches and single-cell omics.

### 4.5. Summary of Achieved In Vivo Studies and Potential Future Directions

This section summarizes the current direct reprogramming approaches for regenerative heart disease treatment based on the cell lines, reprogramming tools, factors (transcription factors or mRNAs), reprogrammed products, administration methods, and therapeutic efficiency, according to availability (Table 1). MI is the most common model used for the examination of direct reprogramming studies. Evaluation of the therapeutic efficiency of direct reprogramming approaches for other heart disease models is significantly lacking.

Furthermore, intracoronary injections and intravenous injections are usually minimally invasive delivery approaches (Figure 4), but have low delivery efficiency due to the complicated physiological filtration system, and are diluted by blood flow. From Table 1 above, intramyocardial injection is the most used administration method, and has been clinically shown to improve retention rate, but requires precise surgical skills to reduce myocardial damage and mechanical injury [117]. Based on clinical stem cell therapy results, intramyocardial injection methods have been considered a safe strategy with better stem cell distribution and retention [118,119]. Moreover, it is also important for reprogramming factors to have a homogeneous distribution in order to prevent potential arrhythmia due to unevenly reprogrammed functional cardiac cell distribution. Li’s group declared that intrapericardial hydrogel injection generates 10-fold higher cell retention and augments the therapeutic effects of MSC in MI over the intramyocardial injection treatment group [120]. This intrapericardial hydrogel method could potentially be applied in order to sustainably localize transcriptional factor release or hiPSC-CMs/iCPCs/iCMs delivery applications, with a larger injection space than the intramyocardial methods. However, both intramyocardial injection and intrapericardial injection require precise surgical skills, which are not always available in developing areas.

Yao’s group has invented an exosome spray that could be applied for acute MI [121]. They fabricated MSC-derived exosomes and biomaterials (such as thrombin), which promoted endogenous angiomyogenesis in the post-injury area of the heart. This technique could be combined with virus-based transcriptional factor delivery by minimizing concerns of random infections. However, in order to apply the spray to the infarcted area, a small surgery is required to avoid pericardial tamponade. Similarly, percutaneous coronary intervention (PCI) surgery could be another practical strategy for reprogramming factor delivery into the infarcted area with little invasive damage [122]. By utilizing sustainable chemical molecule release polymers or hydrogel, it could be precisely delivered into the designated targeted area [123].

Recently, Miragoli et. al. have developed peptide-conjugated biodegradable nanoparticles (cell-penetrating mimetic peptide (R7W-MP), calcium phosphate, size less than 50 nm) which can be delivered into the myocardium through inhalation [124]. However, there remain some major challenges to overcome, as the presence of enzymes in the human airways, such as trypsin, chymotrypsin, and endogenous H_2_O_2_, primarily formed by a source of chronic damage in the aerobic organisms, may disrupt or reduce the therapeutic effect of EVs and other particles. In addition, blood circulation and filtration will further limit efficiency.

In summary, both conventional transplantation and intramyocardial injection are considered invasive clinical approaches. A more reliable, less invasive approach with targeted delivery of transcriptional factors or mRNAs into the infarction area could be a promising breakthrough for improved patient recovery. Furthermore, antigen and antibody-based specific targeting systems could potentially reduce the concern from cytotoxicity and immunology perspectives.

## 5. Conclusions

Cardiomyocyte generation is an essential step in the regenerative medicine model for heart disease repair. The development of iPSCs-derived CMs could potentially provide a powerful and efficient resource for recovering heart function. The direct reprogramming of iCMs or iCPSs from fibroblasts and other somatic cells offers a new perspective for heart disease treatment by converting the fibrosis process into a generation process of functional cardiomyocytes.

Despite several innovative methods that have been developed to promote CM induction, cardiac programming and direct reprogramming still require further investigation. Maturation, immune rejection, engraftment rate, and scalability remain significant barriers to reliable clinical application. A deeper understanding of the cardiac development signal pathway with cross-subject tissue engineering design could enhance and accelerate the translation from laboratory experiments to a clinical setting, potentially saving lives in the process. Long-term safety and therapeutic efficacy also require further exploration. In addition, a combination of gene therapy and cell therapy could be beneficial for regenerative heart disease treatment (Figure 1).

## Figures and Tables

**Figure 1 cells-11-03914-f001:**
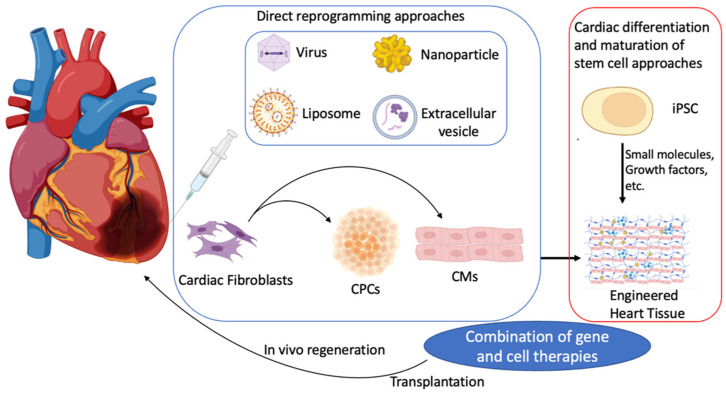
Recent advanced methodologies for regenerative heart disease treatment. hiPSCs differentiate and maturate into hiPSC-CMs for transplantation and further functionalize for engineered heart tissue process (right side). Current genetic direct reprogramming techniques, in vitro and in situ, for cardiac fibroblast cells reprogramming into induced cardiomyocytes (iCMs), and induced cardiac progenitor cells (iCPCs) through transcriptional factors by viruses, microRNAs, and small molecules.

**Figure 2 cells-11-03914-f002:**
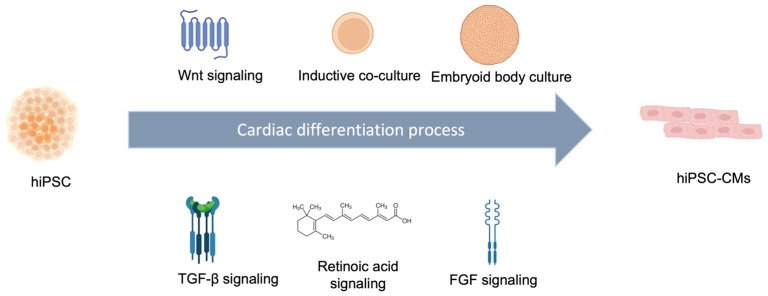
Current development of cardiac differentiation process by hiPSC for cardiomyocyte generation based on Wnt, TGF-β, FGF, and retinoic acid signaling pathway. Culture methods such as inductive co-culture with embryoid body culture.

**Figure 3 cells-11-03914-f003:**
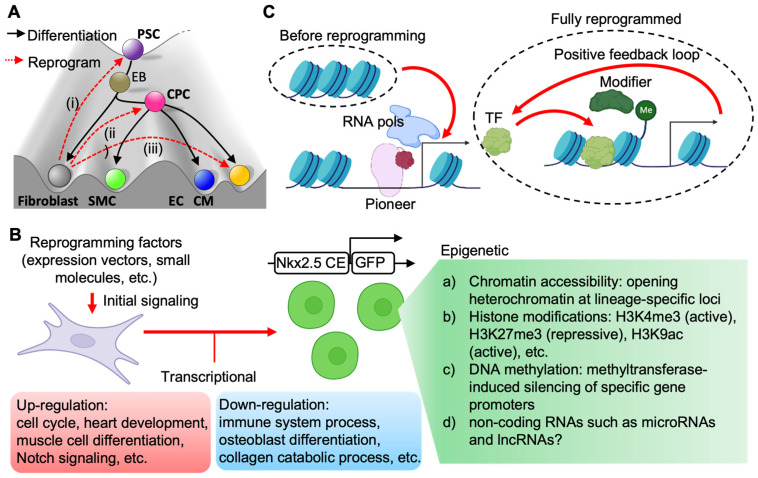
Molecular mechanisms of cardiac reprogramming. (**A**) Various reprogramming routes for converting fibroblasts into cardiovascular cells in Waddington’s epigenetic landscape model. (**B**) The transcriptional and epigenetic mechanisms involved in iCPC reprogramming. (**C**) Potential interactions between reprogramming pioneer factors and epigenetic modifiers, and establishment of an auto-regulatory loop of TFs. PSC: pluripotent stem cells; EB: embryonic body; CPC: cardiovascular progenitor cell; CM: cardiomyocytes; SMC: smooth muscle cell; EC: endothelial cell; TF: transcription factor; CE: cardiac enhancer.

**Figure 4 cells-11-03914-f004:**
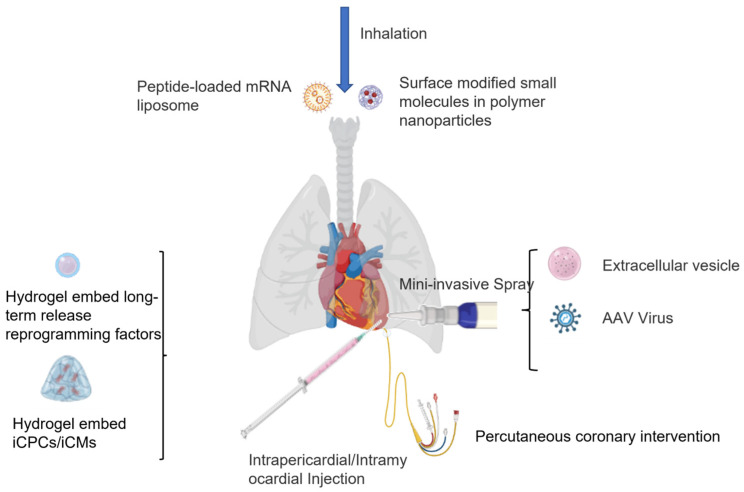
Administration methods for delivery of reprogramming factors and chemical molecules for reprogramming-based therapy.

**Table 1 cells-11-03914-t001:** Summary of achieved in vivo studies for using direct reprogramming techniques for cardiovascular disease treatment.

Cell Lines	Reprogramming Tools	Factors	Reprogramming Efficiency	Administration Method	Therapeutic Effects	Ref.
Mouse cardiac fibroblasts	Sendai Virus (Sev)	GMT	∼40% of iCMs	Sev-GMT Injection	Heart function improvement, little EF change up to 4 weeks (MI) after treatment	[76]
Mouse cardiac fibroblasts	Mesoporous silicon nanoparticles (MSNs)	microRNA1, 133, 208, and 499	iCMs	MSNs mRNA combo heart injection	~50% LEVF after 4 weeks compared to ~25% sham group (MI)	[90]
Smooth muscle cells	Retrovirus	GMT	iCMs	Direct heart virus injection	Scar size decreased from 40% to 20% compared to control MI group	[115]
Mouse cardiac fibroblasts	Cationic gold nanoparticles	GMT	iCMs	Intramyocardial injection	Significant reduction in scar size and fibrosis area	[91]
Mouse embryonic fibroblasts	Extracellular vesicle (EV)	Emb-EV+Mes-EV	~60% of iCMs	Intramyocardial injection	~60% reprogramming efficiency	[101]
Mouse cardiac fibroblasts	Branched polyethyleneimine coated nitrogen-enriched carbon dots	miRNA 1, 133, 208, and 499	iCMs	Myocardium injection	Reduced fibrosis area to approximately 20% under MI disease model	[116]
Mouse tail-tip Fibroblast	Lentivirus	OSKM	iCPCs 90% of engrafted iCPCs efficiently differentiated into CMs, SMC, and EC	Transplantation	Significant small scar sizes after 12 weeks MI surgery	
Human adult dermal fibroblasts	QQ-reagent (a synthesized protein)	GHMT with 3 cytokines	iCPCs	Transplantation	Transplantation into MI rats with fibrosis and LV remodeling decrease	[107]
Adult mouse cardiac fibroblasts	Lentivirus	MTGNB	iCPCs	iCPCs heart injection	~75% survival rate compared to 11% blank control group after 4 weeks (MI)	[73]

## Data Availability

Not applicable.

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
