# Peer review of "Advances in Cellular Reprogramming-Based Approaches for Heart Regenerative Repair"

_cells, 2022, doi:10.3390/cells11233914_

Round 1

Reviewer 1 Report

The manuscript by He et al summarized current advances and challenges in the field of cell reprogramming for heart regeneration. The contents are informative and inspirative. However, there are several questions need to be addressed.

1. Please consider to make the title short and brief. I would suggest to change the title to be: Advances in cellular reprogramming-based approaches for regenerative heart repair.

2. This manuscript needs extensive gramma and phrasing edits. 

3. Lines 465-476, the authors discussed the delivery approaches for the rerogramming treatments, by including intramyocardial and intravenous injection. However, the retention rate is pretty low as regard to the intravenous injections. Given that elevated retention would to be an important indicator for future application, optimized delivery approach, such as intracoronary and intrapericardial injection should be disscussed.

Reviewer 2 Report

The authors aimed to review the recent advances reported on cellular reprogramming- for their application in heart regenerative repair. The study is promising and required in the felid; however, the study missed a lots of recent work (2020-2022) to make connection and highlight the work.  

also, one figure is not enough to show the importance of this work, therefore, more high-quality figure is required to illustrate the cellular and molecular pathway. 
